# Self-cleaving peptides for expression of multiple genes in *Dictyostelium discoideum*

Xinwen Zhu [1,2], Chiara Ricci-Tam [1,2], Emily R. Hager [1,2], Allyson E. Sgro [1,2¤]*

**1** Department of Biomedical Engineering, Boston University, Boston, MA, United States of America,
**2** Biological Design Center, Boston University, Boston, MA, United States of America

¤ Current address: Janelia Research Campus, Howard Hughes Medical Institute, Ashburn, VA, United States of America
* sgroa@janelia.hhmi.org

**Data Availability Statement:** Data and links to the Python code for analysis and re-creating the figures are available on a Dryad repository (DOI: 10.5061/dryad.c866t1g8r). Plasmids for the 2A sequences and the codon-optimized fluorescent

## Abstract

The social amoeba *Dictyostelium discoideum* is a model for a wide range of biological processes including chemotaxis, cell-cell communication, phagocytosis, and development. Interrogating these processes with modern genetic tools often requires the expression of multiple transgenes. While it is possible to transfect multiple transcriptional units, the use of separate promoters and terminators for each gene leads to large plasmid sizes and possible interference between units. In many eukaryotic systems this challenge has been addressed through polycistronic expression mediated by 2A viral peptides, permitting efficient, co-regulated gene expression. Here, we screen the most commonly used 2A peptides, porcine teschovirus-1 2A (P2A), *Thosea asigna* virus 2A (T2A), equine rhinitis A virus 2A (E2A), and foot-and-mouth disease virus 2A (F2A), for activity in *D. discoideum* and find that all the screened 2A sequences are effective. However, combining the coding sequences of two proteins into a single transcript leads to notable strain-dependent decreases in expression level, suggesting additional factors regulate gene expression in *D. discoideum* that merit further investigation. Our results show that P2A is the optimal sequence for polycistronic expression in *D. discoideum*, opening up new possibilities for genetic engineering in this model system.

## Introduction

The social amoeba *Dictyostelium discoideum* is a powerful eukaryotic model system for understanding basic cellular and multicellular behaviors, ranging from chemotaxis and phagocytosis to collective migration and development [1–3]. *D. discoideum's* power as a model system stems from three major sources: First, these behaviors and their molecular drivers are well conserved in other eukaryotic cells such as human cells [4, 5]. Second, it is highly tractable in the lab in both single-cell and multicellular states for experimental manipulation. Finally, *D. discoideum* is easy to genetically manipulate: in addition to having a haploid genome that readily undergoes homologous recombination [6], it can stably maintain extrachromosomal plasmids and express genes from these plasmids [7, 8]. This genetic tractability has been central to its continued importance as a model organism. Increasingly sophisticated genetic manipulations are

proteins created for this study are available from the Dicty Stock Center at http://dictybase.org/StockCenter/StockCenter.html.

**Funding:** "This work was supported by the National Science Foundation grant MCB-1838341 to A.E.S, the National Institutes of Health (NIGMS) grant R35 GM133616 to A.E.S., and the Burroughs Wellcome Fund Career Award at the Scientific Interface to A.E.S. X.Z. was partially supported by the Fonds de recherche du Québec - Nature et technologies (FRQNT). C.R.T. was partially supported by the Biological Design Center Microbiome Initiative Fellowship Program. E.R.H. was partially supported by a Multicellular Design Program fellowship from Boston University's Rajen Kilachand Fund for Integrated Life Sciences and Engineering and is a Simons Foundation Awardee of the Life Sciences Research Foundation. The funders had no role in study design, data collection and analysis, decision to publish, or preparation of the manuscript."

required by modern studies, such as the expression of multiple transgenes simultaneously. Having similar expression levels of these transgenes is often desirable, such as in protein co-localization studies, and in some cases required for optimal function: for example, optogenetic systems often have two components that need to be expressed at similar levels, and biosynthetic pathways for metabolic studies with multiple enzyme components similarly require precise control over relative expression [9, 10].

In *D. discoideum*, transgenes can be expressed either through genomic integration or on extrachromosomally-maintained plasmids. Each approach offers several different ways to express multiple transgenes. In the standard axenic lab strains, known genomic duplications allow for the knock-in of different genes at near-identical genomic loci to achieve similar levels of expression [11]. However, with this approach it must be confirmed that disruption of the chosen loci is not overly deleterious, and a different plasmid is required for each transgene. Transgenes can also be integrated randomly, either separately for each gene or as adjacent transcriptional units on a single plasmid [12]. For applications where clonal populations are not necessary, *D. discoideum* also expresses transgenes from extrachromosomally-maintained plasmids [8, 13]. To express two genes extrachromosomally, they can either be on the same or different plasmids. The two genes have more closely linked expression levels when they are expressed from the same plasmid compared to when they are on separate plasmids [8]. These strategies have been successful, but can have have lower overall transfection efficiencies due to large plasmid size when all the genes are on one plasmid, or because they require either the simultaneous or sequential transfection of multiple plasmids.

A strategy exploited both in nature and in the laboratory to couple expression levels of multiple genes of interest uses a single transcriptional unit for expressing multiple proteins with a polycistronic expression system [14]. This has clear advantages over having a single transcriptional unit code for each product as is currently done in *D. discoideum* monocistronic expression systems. The major advantage is that expression of the gene products in a polycistronic system is co-regulated by the same promoter and terminator, so differential expression and competition for transcription factors is minimized. Having additional gene products controlled by a single promoter and terminator also decreases the required size of transgene expression systems, improving cloning and transfection efficiency [8, 15]. Furthermore, it is possible for transcriptional units that share the same promoter and terminator sequences to undergo homologous recombination, causing one transcriptional unit to substitute itself for the other so only one is present on the plasmid or genome [16], and this possibility is avoided in a polycistronic system.

Polycistronic expression can be achieved using multiple mechanisms, with the most common being viral-derived sequences such as internal ribosome entry site (IRES) sequences or 2A sequences [14]. IRES sequences permit ribosomes to initiate translation at a second open reading frame inside an mRNA strand, and can thus be used to express multiple completely separate proteins. However, IRES sequences are somewhat large, usually over 500 nucleotides in size, and expression of the downstream gene is inefficient and depends on the sequence of the upstream gene [14, 17, 18]. While there are smaller IRES sequences which can have high efficiency when multiple copies are combined in tandem [19], IRES activity also varies greatly depending on cell type, species, and even cell stress state. This is because most IRES sequences require IRES trans-acting factors, in addition to canonical translation initiation factors [20]. In contrast, 2A sequences are under 100 nucleotides in length and some have high efficiency expression of both genes as separate proteins [14]. They also work via a different mechanism from IRES sequences, originally assumed to be and still referred to as "proteolytic cleavage", as first observed between the 2A and 2B regions of the foot-and-mouth disease virus [21]. The true mechanism was later determined to be direct interference with the translational process,

whereby during protein synthesis, a 2A peptide placed immediately upstream of a glycine causes the ribosome to skip the formation of a peptide bond between the glycine and the next amino acid [22, 23]. Multiple 2A-like sequences demonstrating the same ability to separate peptides have been discovered in several different virus families and likely arose independently at multiple points in evolution [24]. This unique mechanism depends mostly on the biochemical properties of a conserved protein complex, the eukaryotic ribosome. As a result, 2A peptides could be expected to be effective in a wide range of eukaryotic cell types. Indeed, 2A peptides have been successfully used for polycistronic expression in a variety of organisms, including animals [25–27], yeast [10], algae [28, 29], and plants [30]. The efficiency of ribosomal skipping for each peptide varies between organisms, necessitating empirical screening to assess which are most appropriate for a given model organism.

Here, we screen the most commonly used 2A peptides, porcine teschovirus-1 2A (P2A), *Thosea asigna* virus 2A (T2A), equine rhinitis A virus 2A (E2A), and foot-and-mouth disease virus 2A (F2A) [31, 32], for activity in *D. discoideum*. We find that all tested peptides are capable of mediating cleavage between an upstream and a downstream peptide, and that P2A constructs have the highest rate of successful ribosomal skipping. We also find that linking two coding sequences lowers overall expression levels, and that the magnitude of this effect differs between two tested strains. These results demonstrate that 2A peptides are functional in *D. discoideum* and that their functionality is highly application-specific.

## Materials and methods

### Molecular cloning

Plasmids used for testing and comparing the different 2A peptides were assembled using the GoldenBraid cloning system, which has been described previously [12]. Each 2A peptide used was codon optimized for expression in *D. discoideum* using the IDT Codon Optimization Tool (Integrated DNA Technologies), synthesized by GenScript, and then cloned into a GoldenBraid pUPD2 vector with the appropriate overhangs to act as a C-terminal linker part. Each transcriptional unit was assembled with an actin15 promoter and an actin8 terminator, and finally assembled into a backbone with a G418-resistance cassette and a *D. discoideum* origin of replication. We used mNeonGreen as a fluorescent protein in all constructs for comparison, mCherry as a red fluorescent protein for all western blotting experiments due to the availability of an antibody with high affinity and specificity, and mScarlet-I as a red fluorescent protein for flow cytometry because it is brighter than mCherry and thus easier to assay [33].

Plasmids for comparing dual cassette and P2A-linked antibiotic resistance gene expression were based on pDM1203 [15]. Note that as a result, all the plasmids for this set of experiments use different actin15 promoter and actin8 terminator sequences derived from this parent plasmid. To make the dual cassette plasmid pDMDC a15-mNG coaA-HygroR, a codon-optimized mNeonGreen was first cloned into the BglII/SpeI cloning site of pDM1203 and the G418-resistance cassette was swapped for a hygromycin-resistance gene by restriction digest with NheI/NotI and Gibson assembly. The hygromygin resistance fragment was obtained using pDM1501 [15] as a PCR template. In contrast to all other plasmids in this study, the resulting dual cassette plasmid is bidirectional, with the antibiotic resistance gene encoded in one direction on the plasmid and the mNeonGreen encoded in the other direction. To make the P2A-linked plasmid pDMP2A a15-mNG-P2A-HygroR, the resistance cassette was removed from pDM1203 by XhoI/BamHI digest followed by blunt-ending and religating the resultant plasmid. A codon-optimized mNeonGreen was then inserted into the BglII/SpeI site along with P2A-HygroR by Gibson assembly. The expression cassettes of the generated plasmids were verified by Sanger sequencing with GENEWIZ (Azenta Life Sciences).

## Cell culture and transfections

AX4 and NC28.1 cells were transfected as previously described [15]. Briefly, cells were grown on a lawn of *Klebsiella aerogenes* on an SM agar plate. The translucent feeding front was harvested and washed once in H40 buffer (40 mM HEPES, 1 mM $MgCl_2$, pH 7.0) by centrifugation at 300g for 3 minutes. Cells were resuspended at $1–4x10^7$ cells/ml in H40 and 100 µl was transferred to a pre-chilled 2 mm gap electroporation cuvette containing 1 µg DNA, and electroporated with a square wave protocol of two pulses of 350 V and 8 milliseconds duration separated by 1 second. 300 µl SorMC (15 mM $KH_2PO_4$, 2 mM $Na_2HPO_4$, 50 µM $MgCl_2$, 50 µM $CaCl_2$) was added to the cells, and 100 µl was transferred to each of 3 wells of a 6 well plate containing 2 ml of *K. aerogenes* suspension (OD600 = 2) in SorMC. After 5 hours recovery, 15 µg/ml G418 or the indicated amount of Hygromycin B Gold (InvivoGen) was added for selection. AX4 cells were left to grow for 3 days before harvesting for fluorescence imaging, flow cytometry, or western blotting, while NC28.1 cells were grown for 2 days. The edges of the plate initiate aggregation by the time of harvest, allowing us to sample a developmentally heterogeneous population.

## Microscopy

For microscopy imaging, cells were washed in Development Buffer (10 mM $K/Na_2$ phosphate buffer, 2 mM $MgSO_4$, and 200 µM $CaCl_2$, pH 6.5) 1–3 times by centrifugation for 3 minutes at 300g to remove bacteria, then placed in a glass bottom dish (Wilco) under a thin (0.3 mm) sheet of 1% agarose. Images were acquired using an inverted epifluorescence microscope (Ti2-E, Nikon) and a silicone-immersion 100X objective (Nikon CFI SR HP Plan Apo Lambda S 100XC Sil, MRD73950). Illumination was provided by a Lumencor Spectra III light engine, set to 50% laser power and attenuated to 5% with an ND 1.3 filter (Chroma UVND_1.3_5%T) in the excitation path, and the camera (Andor iXon Life 888) set to 200 ms exposure. For mNeonGreen-alone expressing cells (S1 Fig), fluorescence was substantially brighter and laser power was set to 5%. Green fluorescence was measured using a 475/28 nm laser (500 mW); 470/20 nm and 570/20 nm double excitation filter (Chroma 59022x); 520/50 nm and 635/80 nm two-band dichroic beamsplitter (Chroma 59022bs); and 525/50 nm emission filter (Chroma ET525/50m). Red fluorescence was measured using a 575/25 nm laser (500 mW); 435/20 nm, 502/20 nm, and 575/20 nm triple excitation filter (Chroma 69008x); 470/30 nm, 540/40 nm, and 635/70 nm three-band dichroic beamsplitter (Chroma 69008bs-uf2); and 632/60 nm emission filter (Chroma ET632/60m).

## Flow cytometry

Cells were grown to near confluency on a 6 well plate (3 days for AX4 or 2 days for NC28.1), then rinsed five times with Development Buffer before being washed off the plate and resuspended in 1 ml Development Buffer. Cells were kept on ice before analysis on an Attune NxT Flow Cytometer (Invitrogen). Green fluorescence was assessed with a blue excitation laser (488 nm, 50 mW) and collecting emitted light with a 530/30 nm filter. Red fluorescence was assessed with a yellow excitation laser (561 nm, 50 mW) and collecting emitted light with a 615/25 nm filter.

Data was analyzed using the Python package FlowCytometryTools [34]. Cells were gated based on forward scatter and side scatter to separate live *D. discoideum* from *K. aerogenes* and from dead cells or cell clumps. Compensation was applied based on spillover coefficients determined from single-color controls, and the cells were then gated into fluorescent and non-fluorescent populations (S2 Fig). For cells determined to have above-threshold fluorescence in

both the green and the red channels, the logarithm of the green-to-red ratio was calculated as:

$$log(green/red) = log(green) - log(red) \tag{1}$$

where log is the natural logarithm. A bias score was calculated for each construct pair by subtracting the mean log(green/red) of the mNeonGreen-first construct from that of the mScarlet-I-first construct.

## Western blotting

Cells grown on a 6 well plate for 3 days were washed in Development Buffer three times by centrifugation for 3 minutes at 300g. The cell pellet was resuspended in 25 μl 2X NuPAGE LDS Sample Buffer (Invitrogen) per harvested well and vortexed to mix. For western blotting, the sample was diluted 1:2 in water and heated at 90°C for 10 minutes. Equal volumes of each sample were run on a 17 well 4–12% Bis-Tris gel at 140V for 50 minutes alongside two molecular weight markers: Flash Protein Ladder 10–180 kDa FPL-006 (Gel Company) in the leftmost lane and PageRuler Plus Prestained Protein Ladder (Thermo Fisher Scientific) in the rightmost lane. Proteins were transferred to a membrane with iBlot at 20V for 7 minutes. Successful transfer was evaluated by labeling the membrane with No-Stain Protein Labeling Reagent (Invitrogen). The membrane was blocked with 5% milk in PBST (bioWORLD) for 1 hour and incubated overnight with primary antibody (Recombinant Anti-mCherry antibody [EPR20579]) diluted 1:5000 in 5% milk in PBST at 4°C. The membrane was then washed 3 times in PBST for 5 minutes each, then incubated in secondary antibody (HRP-conjugated Goat anti-Rabbit IgG, Invitrogen G21234) diluted 1:2000 in 5% milk in PBST for one hour at room temperature. Proteins were detected with SuperSignal West Pico PLUS Chemiluminescent substrate (Thermo Fisher Scientific), and imaged with the iBright FL1000 (Invitrogen). Band intensities were quantified using the iBright Analysis Software.

## Growth and proliferation assay

Cells were seeded in a 6 well plate containing 2 ml of *K. aerogenes* suspension (OD600 = 2) in SorMC and the indicated concentration of Hygromycin B Gold (InvivoGen) at $2x10^5$ cells for AX4 or $1x10^5$ cells for NC28.1. After 24 hours incubation, the bacterial suspension was removed and cells were resuspended in SM media and diluted equally in an overnight culture of *K. aerogenes* grown in SM media such that the concentration of the untreated control was between $2–3x10^6$ cells/ml. A 1:5 serial dilution was performed of each sample in overnight *K. aerogenes* culture and 5 μl was then plated into each well of a 96 well plate that already contained 150 μl of SM agar. Wells were monitored for aggregation and development for two weeks after plating.

## Results and discussion

### 2A viral peptides cleave in *D. discoideum* with different efficiencies

Our approach to expressing multiple genes of interest in *D. discoideum* takes advantage of self-cleaving 2A peptide sequences. To identify 2A peptides with high cleavage efficiencies in *D. discoideum*, we screened the four most commonly used sequences: the porcine teschovirus-1 2A (P2A), *Thosea asigna* virus 2A (T2A), foot-and-mouth disease virus 2A (F2A), and equine rhinitis A virus 2A (E2A) peptides [31, 32]. To increase cleavage efficiency, a glycine-serine-glycine motif was added to the N-terminus of each peptide [35, 36]. Each peptide was also codon optimized for expression in *D. discoideum* (Table 1, S1 Table).

**Table 1. Peptide sequences of 2A viral peptides, along with mean percent cleavage and mean bias score toward expression of the protein ordered first in the coding sequence as measured in this study.**

| 2A | Peptide sequence | % Cleavage | Bias Score |
|---|---|---|---|
| P2A | GSGATNFSLLKQAGDVEENPGP | 99.6% | 0.78 |
| T2A | GSGEGRGSLLTCGDVEENPGP | 99.6% | 1.34 |
| F2A | GSGVKQTLNFDLLKLAGDVESNPGP | 75.7% | - |
| E2A | GSGQCTNYALLKLAGDVESNPGP | 97.7% | - |

2A peptides function by causing the ribosome to skip the formation of a peptide bond at a specific site during translation, creating a second unlinked protein. To compare the ribosomal skipping efficiencies of each peptide, we generated a series of plasmids expressing mCherry-H2B and mNeonGreen separated by either a 2A sequence or by a flexible linker that minimizes repetitive sequences (GSAGSAAGSGEF) [37], as well as a plasmid expressing mNeonGreen alone (S1 Fig). Histone H2B is a well-established nuclear marker, and thus relative subcellular localization of the two fluorophores can be used as a proxy for ribosomal skipping efficiency. In the event of ribosomal skipping, the mCherry fluorophore, being directly linked to H2B, will always remain localized to the nucleus, while the mNeonGreen fluorphore will be able to localize to the cytoplasm. These plasmids were transfected into a standard axenic *D. discoideum* lab strain known as AX4 [38]. AX4 cells transfected with the flexible linker expressed a nuclear-localized mCherry-H2B-mNeonGreen fusion protein. In contrast, cells transfected with the mCherry-H2B-2A-mNeonGreen constructs produced unlinked protein products, which could be visually confirmed by observation of nuclear-localized mCherry and cytoplasmic mNeonGreen with a distribution similar to that of mNeonGreen alone (Fig 1A, S1 Fig). Ribosomal skipping efficiency was quantified by western blot of mCherry (Fig 1B). The full length uncleaved fusion protein has a predicted size of 74 kDa while the 'cleaved' mCherry-H2B fragment has a predicted size of 47 kDa. As expected, the mCherry-H2B-mNeonGreen fusion protein only showed the 74 kDa band while the mCherry-H2B-2A-mNeonGreen constructs had the 47 kDa band and only a faint band at 74 kDa. The secondary bands at approximately 83 kDa and 56 kDa likely correspond to the monoubiquitinated forms of each protein, since monoubiquitination is a known modification of H2B [39]. These secondary bands were excluded from analysis. P2A and T2A demonstrated the most efficient cleavage, both having mean cleavage rates of 99.6%, followed by E2A with 97.7%, and finally F2A with 75.7% (Fig 1C).

## Gene order in the 2A expression cassette affects both relative and overall expression levels

A known drawback of 2A viral peptides is that the ribosome occasionally fails to resume translation after the skipping event, and is instead dissociated from the template in a "fall-off" event. As a result the sequence downstream of the 2A peptide is typically expressed at a lower level than the upstream sequence [40]. To quantify the relative fall-off rates of P2A and T2A sequences and to compare monocistronic and polycistronic expression strategies, we generated a series of constructs that each allow for simultaneous expression of mNeonGreen and mScarlet-I (Fig 2A) and transfected them into AX4. In all constructs, coding sequences for both fluorophores are present on the same plasmid. This single-plasmid approach ensures that in any given cell the two fluorophores are present with the same copy number. In our polycistronic constructs containing dual transcriptional units, both transcriptional units use the same promoter (actin15) and terminator (actin8) as the monocistronic constructs to minimize

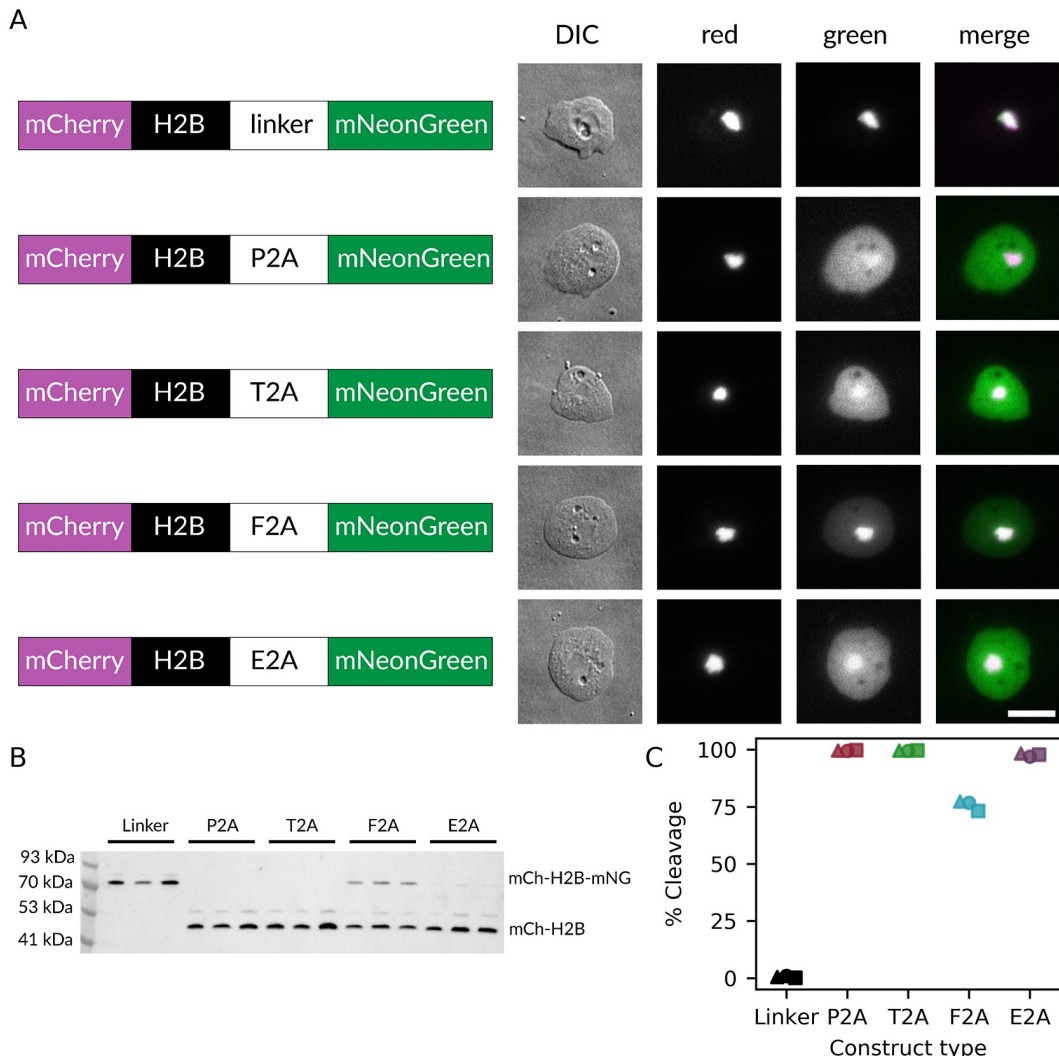

**Fig 1. 2A viral peptides allow for expression of multiple genes in *D. discoideum*.** A: Microscopy images of AX4 *D. discoideum* cells expressing either a directly linked mCherry-H2B-linker-mNeonGreen construct or one of the different mCherry-H2B-2A-mNeonGreen constructs. Representative differential interference contract (DIC), green fluorescence, red fluorescence, and a merged fluorescent image are shown for each construct. Each construct was assembled with an actin15 promoter and an actin8 terminator. Scale bar = 10 μm. B: mCherry western blot of AX4 cells expressing each construct displayed in (A). Blot shows data for three independent transfections. C: Quantification of mCherry western blot. Percent cleavage was calculated as the intensity of the 'cleaved' mCherry-H2B band over the total intensity of the uncleaved and the cleaved bands. Each marker shape indicates an independent experiment (△: replicate 1, ○: replicate 2, □: replicate 3).

expression differences due to variable promoter strength. The green and red fluorescence levels measured by flow cytometry serve as proxies for the expression levels of mNeonGreen and mScarlet-I, and their values relative to each other reveal the relative expression levels of the two fluorescent proteins (Fig 2B). Full details of flow cytometry gating and results are shown in S2–S10 Figs. Both P2A and T2A polycistronic constructs show a clear bias toward the first protein being more highly expressed (Fig 2C). Specifically, the green-to-red fluorescence ratios for both the P2A and the T2A constructs were higher than those of the flexible linker constructs when mNeonGreen was upstream, and lower when mNeonGreen was downstream, consistent with the idea that the upstream protein is expressed in higher quantities than the downstream one. This effect translates into higher bias scores and was more pronounced for

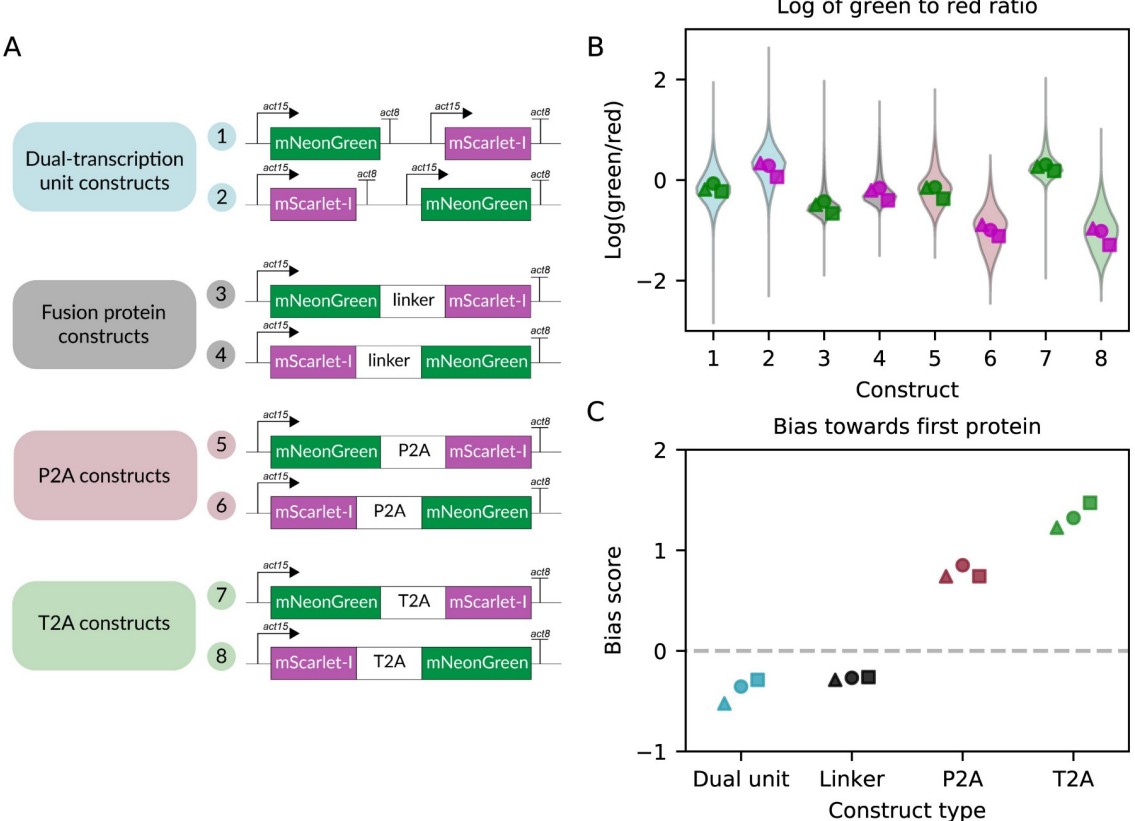

**Fig 2. Order effects are milder for P2A compared to T2A.** A: Constructs transfected into AX4 *D. discoideum* cells. In dual-transcriptional unit constructs, two fluorescent reporters in separate transcriptional units are directly adjacent to each other on the same plasmid. The other construct types co-express both reporters from a single transcriptional unit, joined by either a linker or a 2A peptide sequence. Each transcriptional unit was assembled with an actin15 promoter and an actin8 terminator. B: Natural logarithm of green to red fluorescence ratio calculated for single cells for each construct. The violin plot represents the aggregate single-cell data from three independent transfections, while each individual point is the mean expression level of one transfection. Data below the 0.01th and the 99.99th percentile were excluded from plotting but retained for downstream analysis. Markers filled with green correspond to mNeonGreen-first constructs, and markers filled with magenta correspond to mScarlet-I-first constructs. C: Bias score calculated for each construct type. The bias score is the difference between the mean log(green/red) expression ratio of the mNeonGreen-first construct and that of the mScarlet-I-first construct. In the absence of gene order bias, this score should be zero (dashed line). Positive numbers represent a bias towards the first protein while negative numbers represent a bias towards the second protein. Each marker shape indicates an independent experiment (△: replicate 1, ○: replicate 2, □: replicate 3).

T2A than for P2A, indicating that successful resumption of translation occurs less often with T2A than with P2A. Based on these findings, we recommend P2A as the optimal 2A sequence among those tested for use in *D. discoideum*.

Beyond this expected bias toward the first protein for sequences using 2A viral peptides, we also observed several other unexpected effects. First, both the direct linker and the dual cassette constructs display a bias towards the second protein (Fig 2C). In the case of the linker constructs, it is possible that the folding of the first protein was disrupted. For the dual cassette constructs, the mechanism is unclear, but similar results have been observed in other systems [41].

To investigate overall expression, rather than relative expression, we next considered the mean green and red fluorescence values separately (Fig 3A and 3B). The relative positioning of mNeonGreen and mScarlet-I had unexpected effects on overall expression levels: the fusion protein and polycistronic 2A constructs had higher overall fluorescence in both green and red

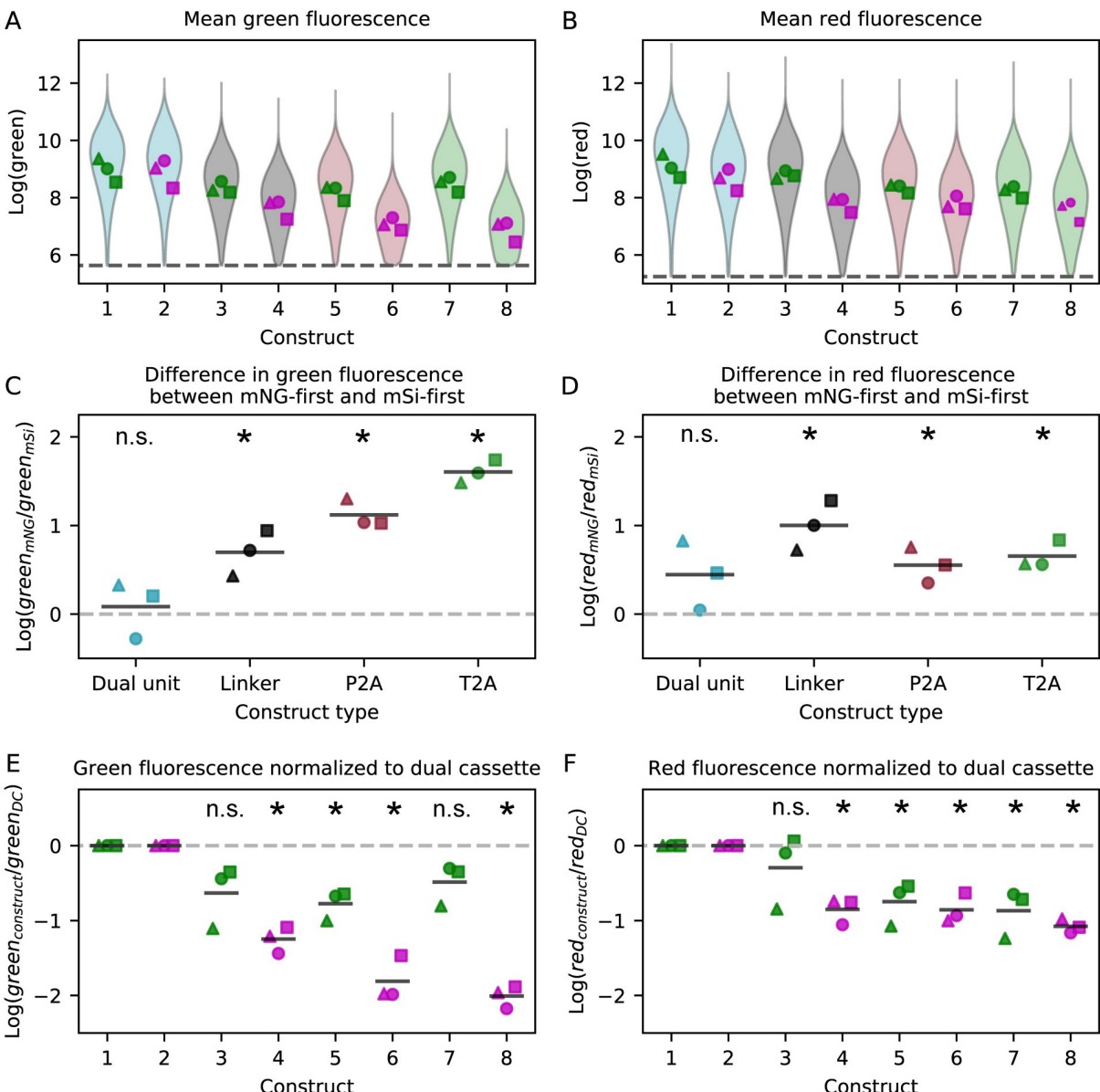

**Fig 3. Expression level differences between constructs.** A and B: Mean green (A) and mean red (B) expression levels for each construct measured by flow cytometry. The violin plot represents the aggregate single-cell data from three independent transfections, while each individual point is the mean expression level of one transfection. Markers filled with green correspond to mNeonGreen-first constructs, and markers filled with magenta correspond to mScarlet-I-first constructs. Dashed lines represent fluorescence thresholds on which data was gated (S2 Fig). C and D: Difference in mean green (C) and mean red (D) expression between the mNeonGreen-first construct and the mScarlet-I-first construct for each construct type. Solid horizontal lines indicate the mean of each construct. Two-tailed t-tests determined if the means were statistically different from zero (dashed line), n.s.: not significant, *: $p < 0.05$. E and F: Mean green (E) and red (F) fluorescence values for each construct normalized to the dual cassette construct with the same ordering. Solid horizontal lines indicate the mean of each construct. Two-tailed t-tests determined if means were statistically different from zero (dashed line), n.s.: not significant, *: $p < 0.05$. Each marker shape indicates an independent experiment (△: replicate 1, ○: replicate 2, □: replicate 3).

when the mNeonGreen was upstream, as determined by calculating the difference in fluorescence between the mNeonGreen-first and the mScarlet-I-first constructs (Fig 3C and 3D). This suggests that part of the mNeonGreen coding sequence may promote higher overall expression when it is near the start of a transcript. This idea that coding sequences affect

expression is consistent with the observation that the monocistronic vectors both had higher green-to-red fluorescence ratios than the fusion protein constructs (Fig 2B), since this could also be explained by the mNeonGreen being preferentially expressed over mScarlet-I despite both transcriptional units using the same promoter sequence. While both our mNeonGreen and mScarlet-I coding sequences are codon optimized for *D. discoideum*, our results may also suggest that the mScarlet-I coding sequence could be further optimized or that there are other post-transcriptional effects regulating expression of the mScarlet-I-first constructs.

The linked fusion protein and 2A constructs also both display lower overall expression levels compared to the dual cassette constructs, an effect that is not statistically significant for all mNeonGreen-first constructs but very clear for the lower-expressing mScarlet-I-first constructs (Fig 3E and 3F). One possible explanation for this effect is that longer transcripts are not expressed efficiently. The higher overall expression from the dual expression cassette is particularly notable given that the plasmid constructs are 7–8% larger than the fusion protein and 2A constructs. We used a constant mass of DNA per transfection, 1 µg, so the number of molecules of dual cassette plasmid transfected was only 93% of that of the fusion protein and 2A plasmids. Furthermore, larger plasmids transfect with a lower frequency [8, 15], further supporting the idea that longer transcripts may be expressed less efficiently than multiple smaller ones. This result also suggests that the number of transcription factors available to initiate transcription is not the rate-limiting step for transient expression in *D. discoideum*, because if transcription factors were limiting, competition between identical promoters would be expected to reduce expression of both genes.

## Using 2A peptides to link a gene of interest to antibiotic resistance results in lower expression

Viral 2A peptides have been used to link antibiotic expression genes downstream of a gene of interest to force expression of that gene in all antibiotic-resistant cells [42]. We tested this strategy by placing an mNeonGreen coding sequence upstream of a P2A and an antibiotic resistance gene. However, during preliminary trials in AX4, transfection of a plasmid with the G418 resistance gene downstream of P2A yielded only a single transfectant which took a week to expand to fill a well in a six-well plate. In contrast, transient transfection with G418 selection plasmids under these conditions typically yields such a large number of colonies that they cannot be easily distinguished, and the plates reach confluency at 3 days post-transfection. Previous literature suggests that integrating G418 plasmids are maintained at very high copy number in *D. discoideum* [43], suggesting that the resistance gene must be expressed at high levels to be effective. Our results demonstrate that in *D. discoideum*, like in other systems [40], genes downstream of P2A are expressed less efficiently than genes in dual cassette systems (Fig 2). As a result, it is possible that the G418 resistance gene could not be expressed at high enough levels to effectively confer resistance when positioned after the P2A sequence, which resulted in the observed absence of growth.

In contrast, hygromycin resistance plasmids are normally maintained at intermediate copy numbers in integrating expression systems [43]. While untargeted genomic integration events are random and thus the number of copies is also random, the copy number of extrachromosomal plasmids is expected to be determined by the replication sequence. Our plasmids retain the Ddp1-derived replication system from the pDM series of *D. discoideum* expression constructs [8]. While the copy numbers of these constructs have not been evaluated, Ddp1 has previously been estimated to be present at 100 copies per cell [44]. In both the genomic integration and the extrachromosomal case, cells with inadequate expression will be selected against by the presence of antibiotic, and the observed population will skew towards cells with

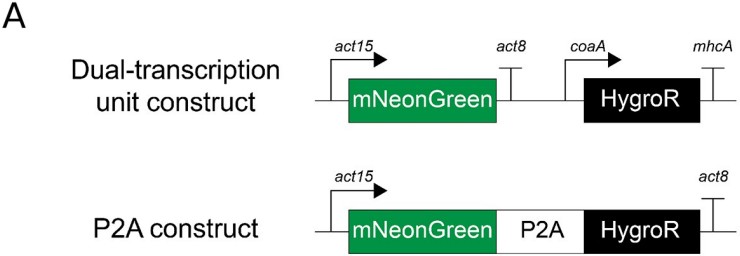

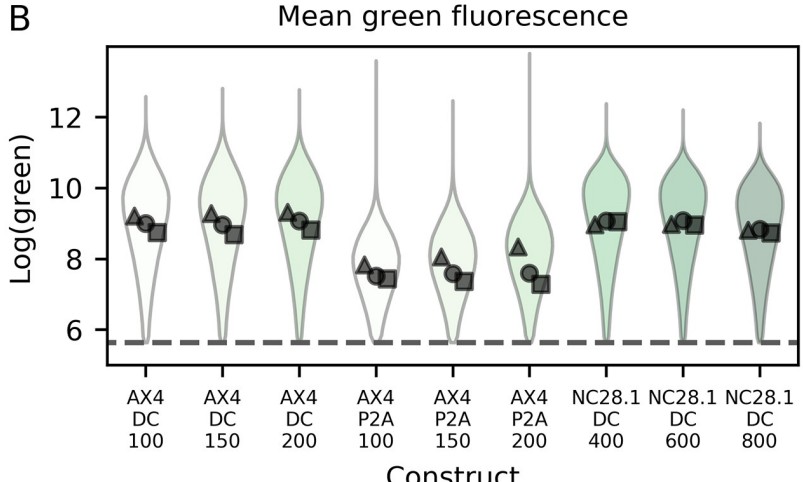

**Fig 4. Linking gene of interest to hygromycin expression cassette lowers overall expression.** A: Constructs transfected into *D. discoideum* cells. In the dual-transcriptional unit construct the mNeonGreen is driven by an actin15 promoter and has an actin8 terminator while the hygromycin resistance gene is driven by a coaA promoter and has a mhcA terminator. In the P2A construct, mNeonGreen-P2A-HygroR is driven by an actin15 promoter and has an actin8 terminator. B: Mean green expression in either AX4 or NC28.1 measured by flow cytometry. The violin plot represents the aggregate single-cell data from three independent transfections, while each individual point is the mean expression level of one transfection. Dashed lines represent fluorescence thresholds. X-axis labels indicate whether the dual cassette (DC) or the P2A construct was used, alongside the hygromycin concentration in μg/ml. Each marker shape indicates an independent experiment (△: replicate 1, ○: replicate 2, □: replicate 3).

higher copy numbers. Previous work suggests that within a limited range, increasing the antibiotic concentration used for selection of a transiently-transfected population will skew the population mean towards higher expression levels [15]. To explore this effect, we chose to test a range of hygromycin concentrations for use to see if hygromycin resistance could be expressed at high enough levels when following a P2A sequence to maintain a population with high copy numbers of plasmids and also drive higher expression of a gene of interest.

We compared a dual-cassette plasmid, expressing mNeonGreen and the hygromycin resistance gene as separate transcriptional units, to one where the two coding sequences are linked by a P2A sequence. Each was transfected into both AX4 and the natural isolate strain NC28.1 [45] (Fig 4A). These two strains differ in their susceptibility to hygromycin (S11 Fig), so a different range of Hygromycin B Gold concentrations was used for each. Full details of flow cytometry results are shown in S12–S23 Figs. In AX4, the mNeonGreen expression was lower but still present in the cells using P2A for expression of both genes (Fig 4B). In NC28.1, the cells transfected with separate transcriptional units grew well, indicating expression of the antibiotic resistance gene, and had strong mNeonGreen expression (Fig 4B). However, NC28.1 cells transfected with the P2A construct had lower overall cell counts under the same growth

conditions and never more than 6% of cells met the fluorescence threshold for quantification in any replicate for any antibiotic condition (S21–S23 Figs). This suggests that with a P2A sequence neither the antibiotic resistance nor the mNeonGreen genes were well-expressed in NC28.1. No clear trend between antibiotic concentration and expression level can be seen in our results.

## Conclusion

In this work, we demonstrate that viral 2A sequences can mediate ribosomal skipping in *D. discoideum*. Of the common sequences tested, P2A performs the best both in terms of highest skipping rates (highest proportion of proteins separated) and successful resumption of translation (least bias towards first protein). Our results also illustrate that longer transcripts containing multiple protein coding sequences, either fused into a single protein with a linker or designed to be produced as two separate proteins with a 2A sequence, have lower expression of both genes and the strength of this effect differs between the standard lab strain and a natural isolate strain. Previous work suggests that in systems with low transcription factor concentrations, these low levels rate-limit transcription initiation and the resulting competition between promoters for transcription factors dominates gene expression patterns [46, 47]. Under this scenario, we would have expected to see the opposite effect on expression level: putting two fluorophores under a single promoter instead of two identical promoters competing for transcription factor binding would have increased production of both. Instead, we observed a decrease in overall expression level with longer transcript lengths, suggesting that transcription initiation factors are in excess in *D. discoideum*, and that the rate-limiting step in gene expression lies further downstream of transcription initiation.

The 2A sequences used in this work were codon optimized for expression in *D. discoideum*. The purpose of codon optimization is to increase translational efficiency of a heterologous peptide by replacing the coding sequence with synonymous codons towards which the host organism displays bias [48, 49]. While codon optimization is routinely used for heterologous expression [50], in the case of 2A sequences, wild-type viral sequences may be evolutionarily optimized to support the cleavage mechanism [51]. It is therefore unclear whether codon optimization is the best strategy for 2A peptides, and future studies could explore the effects of alternative codon usage on cleavage rates.

The functionality of viral 2A peptides in *D. discoideum* opens up a new route to expressing multiple genes of interest at similar levels in fewer steps than some other approaches. This may be of particular interest for applications that require coordinated stoichiometric expression. For instance, two-part optogenetic systems, such as those based on CRY2/CIBN interactions, are of interest in *D. discoideum* [52]. These systems have benefited from concerted expression of the two interacting proteins using a 2A peptide in other model organisms [9]. A recently-published study using *D. discoideum* as a chassis for the production of polyketides [53] accomplished polycistronic expression of multiple enzymes with a P2A-based expression system originally developed for fungi [54], demonstrating the utility of closely linked expression levels in *D. discoideum* metabolic engineering. Furthermore, reducing the number of identical promoters and terminators used to co-transfect multiple genes could decrease the risk of unwanted recombination events [16]. As a result, we anticipate 2A peptides, and P2A in particular, will be a useful addition to the *D. discoideum* genetic toolkit.

## Supporting information

**S1 Raw images. Raw images of western blots.**
(PDF)

**S1 Table. DNA sequences of 2A viral peptides.**
(XLSX)

**S2 Table. Table of flow cytometry experiments.**
(XLSX)

**S1 Fig. Cells expressing mNeonGreen alone.** Microscopy images of AX4 *D. discoideum* cells under differential interference contrast microscopy (left, DIC) and green fluorescence (right). Each cell is taken from a different field of view, from one of three independent transfections (top two rows: replicate 1, middle two rows: replicate 2, bottom two rows: replicate 3). Color bars show arbitrary intensity units in each channel. Each field of view is 26 μm x 26 μm.
(TIFF)

**S2 Fig. Gating and spillover compensation for flow cytometry.** A-C: Forward scatter and side scatter profile of *K. aerogenes* used as a food source for cells post-transfection (A), AX4 cells mock transfected without DNA and incubated with 15 μg/ml G418 for three days (B), and AX4 harvested from a lawn of *Klebsiella aerogenes* on an SM agar plate (C). The black polygon represents the gate separating live Dicty from bacteria, dead cells, or other debris, and only events falling inside the polygon gate were considered for downstream analysis. D-E: Fluorescence values for AX4 cells transfected with mNeonGreen only (D) or mScarlet-I only (E). Spillover coefficients for each fluorescent protein were determined by linear fitting. F-I: Log-transformed uncompensated fluorescence values for AX4 cells that are untransfected (F), transfected with mNeonGreen only (G), mScarlet-I only (H), or with a co-expression construct (I). The horizontal and vertical lines represent the fluorescence thresholds for green and red, respectively. J-M: Fluorescence values from (F-I) after application of the appropriate spillover compensations. Note that J is identical to F because no compensation was required.
(TIFF)

**S3 Fig. Flow cytometry data of AX4 cells transfected with dual transcription cassettes for mNeonGreen and mScarlet-I.** Top: Forward scatter and side scatter profiles of all three replicates. Density map color indicates event count. Events inside the polygon, defined as shown in S2 Fig, were considered live cells. Middle: Green and red fluorescence values for live cells for each replicate. Displayed numbers indicate total event count in a given quadrant. Cells in the upper right quadrant were retained for further analysis. Bottom: Mean fluorescence values and ratio of green to red fluorescence for each replicate. For each violin plot of single-cell values, the overlaid dot represents mean of the distribution and the error bars represent the standard deviation. Dashed lines represent fluorescence thresholds.
(TIFF)

**S4 Fig. Flow cytometry data of AX4 cells transfected with dual transcription cassettes for mScarlet-I and mNeonGreen.** Top: Forward scatter and side scatter profiles of all three replicates. Density map color indicates event count. Events inside the polygon, defined as shown in S2 Fig, were considered live cells. Middle: Green and red fluorescence values for live cells for each replicate. Displayed numbers indicate total event count in a given quadrant. Cells in the upper right quadrant were retained for further analysis. Bottom: Mean fluorescence values and ratio of green to red fluorescence for each replicate. For each violin plot of single-cell values, the overlaid dot represents mean of the distribution and the error bars represent the standard deviation. Dashed lines represent fluorescence thresholds.
(TIFF)

**S5 Fig. Flow cytometry data of AX4 cells transfected with mNeonGreen-linker-mScarlet-I.** Top: Forward scatter and side scatter profiles of all three replicates. Density map color

indicates event count. Events inside the polygon, defined as shown in S2 Fig, were considered live cells. Middle: Green and red fluorescence values for live cells for each replicate. Displayed numbers indicate total event count in a given quadrant. Cells in the upper right quadrant were retained for further analysis. Bottom: Mean fluorescence values and ratio of green to red fluorescence for each replicate. For each violin plot of single-cell values, the overlaid dot represents mean of the distribution and the error bars represent the standard deviation. Dashed lines represent fluorescence thresholds.
(TIFF)

**S6 Fig. Flow cytometry data of AX4 cells transfected with mScarlet-I-linker-mNeonGreen.**
Top: Forward scatter and side scatter profiles of all three replicates. Density map color indicates event count. Events inside the polygon, defined as shown in S2 Fig, were considered live cells. Middle: Green and red fluorescence values for live cells for each replicate. Displayed numbers indicate total event count in a given quadrant. Cells in the upper right quadrant were retained for further analysis. Bottom: Mean fluorescence values and ratio of green to red fluorescence for each replicate. For each violin plot of single-cell values, the overlaid dot represents mean of the distribution and the error bars represent the standard deviation. Dashed lines represent fluorescence thresholds.
(TIFF)

**S7 Fig. Flow cytometry data of AX4 cells transfected with mNeonGreen-P2A-mScarlet-I.**
Top: Forward scatter and side scatter profiles of all three replicates. Density map color indicates event count. Events inside the polygon, defined as shown in S2 Fig, were considered live cells. Middle: Green and red fluorescence values for live cells for each replicate. Displayed numbers indicate total event count in a given quadrant. Cells in the upper right quadrant were retained for further analysis. Bottom: Mean fluorescence values and ratio of green to red fluorescence for each replicate. For each violin plot of single-cell values, the overlaid dot represents mean of the distribution and the error bars represent the standard deviation. Dashed lines represent fluorescence thresholds.
(TIFF)

**S8 Fig. Flow cytometry data of AX4 cells transfected with mScarlet-I-P2A-mNeonGreen.**
Top: Forward scatter and side scatter profiles of all three replicates. Density map color indicates event count. Events inside the polygon, defined as shown in S2 Fig, were considered live cells. Middle: Green and red fluorescence values for live cells for each replicate. Displayed numbers indicate total event count in a given quadrant. Cells in the upper right quadrant were retained for further analysis. Bottom: Mean fluorescence values and ratio of green to red fluorescence for each replicate. For each violin plot of single-cell values, the overlaid dot represents mean of the distribution and the error bars represent the standard deviation. Dashed lines represent fluorescence thresholds.
(TIFF)

**S9 Fig. Flow cytometry data of AX4 cells transfected with mNeonGreen-T2A-mScarlet-I.**
Top: Forward scatter and side scatter profiles of all three replicates. Density map color indicates event count. Events inside the polygon, defined as shown in S2 Fig, were considered live cells. Middle: Green and red fluorescence values for live cells for each replicate. Displayed numbers indicate total event count in a given quadrant. Cells in the upper right quadrant were retained for further analysis. Bottom: Mean fluorescence values and ratio of green to red fluorescence for each replicate. For each violin plot of single-cell values, the overlaid dot represents mean of the distribution and the error bars represent the standard deviation. Dashed lines

represent fluorescence thresholds.
(TIFF)

**S10 Fig. Flow cytometry data of AX4 cells transfected with mScarlet-I-T2A-mNeonGreen.**
Top: Forward scatter and side scatter profiles of all three replicates. Density map color indicates event count. Events inside the polygon, defined as shown in S2 Fig, were considered live cells. Middle: Green and red fluorescence values for live cells for each replicate. Displayed numbers indicate total event count in a given quadrant. Cells in the upper right quadrant were retained for further analysis. Bottom: Mean fluorescence values and ratio of green to red fluorescence for each replicate. For each violin plot of single-cell values, the overlaid dot represents mean of the distribution and the error bars represent the standard deviation. Dashed lines represent fluorescence thresholds.
(TIFF)

**S11 Fig. Assay of growth and development after treatment with hygromycin.** A: Diagram illustrating assay used to determine antibiotic susceptibility for different cell lines. Cells were grown with the indicated concentration of Hygromycin B Gold for 24 hours, then diluted and plated on SM agar. Wells were monitored for aggregation and development for two weeks after plating. At right, the final growth outcomes are shown for each replicate. B-C: Summarized assay results for AX4 (B) and NC28.1 (C). Each marker shows an independent experiment.
(TIF)

**S12 Fig. Flow cytometry data of AX4 cells transfected with the dual cassette plasmid and selected with 100 μg/ml hygromycin.** Top: Forward scatter and side scatter profiles of all three replicates. Events inside the polygon were considered live cells. Middle: Green fluorescence values for live cells for each replicate. Displayed numbers indicate the total event count and the percentage of events below or above the threshold. Cells above the fluorescence threshold were retained for further analysis. Bottom: Mean fluorescence values for each replicate.
(TIFF)

**S13 Fig. Flow cytometry data of AX4 cells transfected with the dual cassette plasmid and selected with 150 μg/ml hygromycin.** Top: Forward scatter and side scatter profiles of all three replicates. Events inside the polygon were considered live cells. Middle: Green fluorescence values for live cells for each replicate. Displayed numbers indicate the total event count and the percentage of events below or above the threshold. Cells above the fluorescence threshold were retained for further analysis. Bottom: Mean fluorescence values for each replicate.
(TIFF)

**S14 Fig. Flow cytometry data of AX4 cells transfected with the dual cassette plasmid and selected with 200 μg/ml hygromycin.** Top: Forward scatter and side scatter profiles of all three replicates. Events inside the polygon were considered live cells. Middle: Green fluorescence values for live cells for each replicate. Displayed numbers indicate the total event count and the percentage of events below or above the threshold. Cells above the fluorescence threshold were retained for further analysis. Bottom: Mean fluorescence values for each replicate.
(TIFF)

**S15 Fig. Flow cytometry data of AX4 cells transfected with the P2A plasmid and selected with 100 μg/ml hygromycin.** Top: Forward scatter and side scatter profiles of all three replicates. Events inside the polygon were considered live cells. Middle: Green fluorescence values for live cells for each replicate. Displayed numbers indicate the total event count and the percentage of events below or above the threshold. Cells above the fluorescence threshold were

retained for further analysis. Bottom: Mean fluorescence values for each replicate.
(TIFF)

**S16 Fig. Flow cytometry data of AX4 cells transfected with the P2A plasmid and selected with 150 μg/ml hygromycin.** Top: Forward scatter and side scatter profiles of all three replicates. Events inside the polygon were considered live cells. Middle: Green fluorescence values for live cells for each replicate. Displayed numbers indicate the total event count and the percentage of events below or above the threshold. Cells above the fluorescence threshold were retained for further analysis. Bottom: Mean fluorescence values for each replicate.
(TIFF)

**S17 Fig. Flow cytometry data of AX4 cells transfected with the P2A plasmid and selected with 200 μg/ml hygromycin.** Top: Forward scatter and side scatter profiles of all three replicates. Events inside the polygon were considered live cells. Middle: Green fluorescence values for live cells for each replicate. Displayed numbers indicate the total event count and the percentage of events below or above the threshold. Cells above the fluorescence threshold were retained for further analysis. Bottom: Mean fluorescence values for each replicate.
(TIFF)

**S18 Fig. Flow cytometry data of NC28.1 cells transfected with the dual cassette plasmid and selected with 400 μg/ml hygromycin.** Top: Forward scatter and side scatter profiles of all three replicates. Events inside the polygon were considered live cells. Middle: Green fluorescence values for live cells for each replicate. Displayed numbers indicate the total event count and the percentage of events below or above the threshold. Cells above the fluorescence threshold were retained for further analysis. Bottom: Mean fluorescence values for each replicate.
(TIFF)

**S19 Fig. Flow cytometry data of NC28.1 cells transfected with the dual cassette plasmid and selected with 600 μg/ml hygromycin.** Top: Forward scatter and side scatter profiles of all three replicates. Events inside the polygon were considered live cells. Middle: Green fluorescence values for live cells for each replicate. Displayed numbers indicate the total event count and the percentage of events below or above the threshold. Cells above the fluorescence threshold were retained for further analysis. Bottom: Mean fluorescence values for each replicate.
(TIFF)

**S20 Fig. Flow cytometry data of NC28.1 cells transfected with the dual cassette plasmid and selected with 800 μg/ml hygromycin.** Top: Forward scatter and side scatter profiles of all three replicates. Events inside the polygon were considered live cells. Middle: Green fluorescence values for live cells for each replicate. Displayed numbers indicate the total event count and the percentage of events below or above the threshold. Cells above the fluorescence threshold were retained for further analysis. Bottom: Mean fluorescence values for each replicate.
(TIFF)

**S21 Fig. Flow cytometry data of NC28.1 cells transfected with the P2A plasmid and selected with 400 μg/ml hygromycin.** Top: Forward scatter and side scatter profiles of all three replicates. Events inside the polygon were considered live cells. Middle: Green fluorescence values for live cells for each replicate. Displayed numbers indicate the total event count and the percentage of events below or above the threshold. Cells above the fluorescence threshold were retained for further analysis. Bottom: Mean fluorescence values for each replicate.
(TIFF)

**S22 Fig. Flow cytometry data of NC28.1 cells transfected with the P2A plasmid and selected with 600 μg/ml hygromycin.** Top: Forward scatter and side scatter profiles of all three replicates. Events inside the polygon were considered live cells. Middle: Green fluorescence values for live cells for each replicate. Displayed numbers indicate the total event count and the percentage of events below or above the threshold. Cells above the fluorescence threshold were retained for further analysis. Bottom: Mean fluorescence values for each replicate. (TIFF)

**S23 Fig. Flow cytometry data of NC28.1 cells transfected with the P2A plasmid and selected with 800 μg/ml hygromycin.** Top: Forward scatter and side scatter profiles of all three replicates. Events inside the polygon were considered live cells. Middle: Green fluorescence values for live cells for each replicate. Displayed numbers indicate the total event count and the percentage of events below or above the threshold. Cells above the fluorescence threshold were retained for further analysis. Bottom: Mean fluorescence values for each replicate. (TIFF)

## Acknowledgments

The authors thank Dr. Alison Tebo and Dr. Liana Lareau for their critical review of our work. We also thank the members of the Sgro Lab for feedback throughout the project.

## Author Contributions

**Conceptualization:** Xinwen Zhu, Chiara Ricci-Tam, Allyson E. Sgro.

**Funding acquisition:** Allyson E. Sgro.

**Investigation:** Xinwen Zhu, Chiara Ricci-Tam, Emily R. Hager.

**Methodology:** Xinwen Zhu, Chiara Ricci-Tam, Emily R. Hager, Allyson E. Sgro.

**Supervision:** Allyson E. Sgro.

**Visualization:** Xinwen Zhu, Chiara Ricci-Tam, Emily R. Hager, Allyson E. Sgro.

**Writing – original draft:** Xinwen Zhu, Chiara Ricci-Tam, Allyson E. Sgro.

**Writing – review & editing:** Xinwen Zhu, Chiara Ricci-Tam, Emily R. Hager, Allyson E. Sgro.

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
