## [Decision Letter · Decision Letter 0]

1 May 2022

PONE-D-22-08221Self-cleaving peptides for expression of multiple genes in Dictyostelium discoideumPLOS ONE

Dear Dr. Allyson E. Sgro,

Thank you for submitting your manuscript to PLOS ONE. After careful consideration, we feel that it has merit but does not fully meet PLOS ONE’s publication criteria as it currently stands. Therefore, we invite you to submit a revised version of the manuscript that convincingly addresses the points and questions raised by both constructive reviewers.

We look forward to receiving your revised manuscript.

Kind regards,

Maria Gasset, Ph.D.

Academic Editor

PLOS ONE

Journal Requirements:

"The authors thank Dr. Alison Tebo and Dr. Liana Lareau for their critical review of our work. This work was supported by the National Science Foundation grant MCB-1838341 to A.E.S and the Burroughs Wellcome Fund Career Award at the Scientific Interface to A.E.S. X.Z. was partially supported by the Fonds de recherche du Qu´ebec - Nature et technologies (FRQNT) and C.R.T. was partially supported by the

Biological Design Center Microbiome Initiative Fellowship Program"

Reviewers' comments:

Reviewer's Responses to Questions

**Comments to the Author**

1. Is the manuscript technically sound, and do the data support the conclusions?

Reviewer #1: Yes

Reviewer #2: Partly

2. Has the statistical analysis been performed appropriately and rigorously? 

Reviewer #1: Yes

Reviewer #2: Yes

3. Have the authors made all data underlying the findings in their manuscript fully available?

Reviewer #1: Yes

Reviewer #2: Yes

4. Is the manuscript presented in an intelligible fashion and written in standard English?

Reviewer #1: Yes

Reviewer #2: Yes

5. Review Comments to the Author

Reviewer #1: The submitted work systematically establishes the usage of 2A peptides for the expression of dual reporters in Dictyostelium.

I deeply appreciate the effort the authors put into this manuscript to provide another useful tool for exploring different aspects of biology in Dictyostelium cells.

Well done on the statistics and flow cytometry data - that was a lot of work.

My comments and questions to the authors:

Figure 1:

Did you perform any protein level quantification before your loaded you gel to ensure equal amounts of protein was analyzed for all five constructs in your blot (BCA, Bradford or Precision Red - have not found it in the methods)? Or maybe you have a loading control like Tubulin? It would be a nice addition if you have the data already and just not shown in your manuscript since you do not use it for your quantification. It would further support your statement - at the moment it could also be that P2A looks better than T2A because you have loaded less sample.

Figure 2/3:

How do your dual reporters work? Are they expressed from the same plasmid? Are both constructs (mCherry and mNeonGreen) using an act15 promoter and an act8 terminator?

It was not entirely clear to me from the text. Maybe you could explain this briefly in the main text or the figure legends?

Do you think that maturation effects (10 min mNeon and 36 min mScarlet-I) could partially explain the better expression of constructs where mNeon is first? I always assumed that the oxidation step for the folding of fluorescent proteins makes the difference but you data might indicate that this is not the whole story.

Figure 4:

It might be a naive question, but I am not entirely sure after reading the text and figure legend: Is the construct driven by the act15 promoter or the CoA promoter? From our experience the CoA promoter is able to deliver enough protein to give cells G418 resistance under non-axenic conditions (We use it for knock outs and knock ins). We never texted act15 in this context, so I started wondering.

It is remarkable how resistant your NC4 isolate is towards hygromycin for DC conditions. Have you tested lower amounts of antibiotic for the screening when using the P2A construct to maybe obtain some clones?

Have you tested another non-axenic or axenic cell line? Or do you think the effect you see is unrelated to the ability to live in bacteria free liquid medium?

Three general question:

If I am correct you have always used extrachromosomal plasmids for your studies. Have you ever tried P2A or T2A in an integrating system like REMI or Act5? I was wondering if those strategies which depend on low copies of construct could also benefit from the 2A system or if the the decrease in expression you saw for the extrachromosomal plasmids would require further optimization for those (Please don't test it - I just ask out of curiosity).

Did you ever tested more than 2 reporters at the same time? I am just wondering if ribosomal skipping is getting sequentially worse? (Please don't test this either)

Have you tested different promoters and terminators or did you directly decided on act15 and act8? I was just wondering about the effects of different transcription initiators and terminators (Curiosity again) in terms of expression drop when using 2A sequences.

Good luck for the final submission!

Reviewer #2: The present manuscript by Sgro and colleagues is a technical report of the establishment and quantification of polycistronic expression for the model organism Dictyostelium. They demonstrate that 2A viral peptides are capable of mediating effective cleavage between first and second protein and that gene order effects are milder for P2A. This is an interesting study that sheds further light into the genetic toolkit of Dictyostelium. These data are generally of high quality, and the manuscript is within the remit of PLoS One. However, there are several concerns with the interpretation of the data.

Major comments:

1- The authors show convincing evidence that 2A peptides are capable of producing unlinked protein products. However, even with P2A, which demonstrated the highest 99.6% of cleavage efficiency, mNG fluorescence is enriched at centre of the cell enough to identify nuclear shape. Therefore, it seems that mNG data do not fully support the conclusion. Especially, it is possible that the present cleavage efficiencies do not reflect the dynamic behaviour of the protein in the cells. The recent manuscript presented by Hillmann and colleagues showed viral 2A sequences function in the expression of multiple enzymes in Dictyostelium, thus precise quantification of the efficiency is an important piece in this study.

2- Related to the above point, it would be good to quantify the nuclear to cytoplasmic ration of mNG using the imaging data. What about the impact of cleavage efficiency calculated by the different approaches? The authors should also show whether mNG without H2B distributes uniformly throughout the cell. Might be enough to be shown in the Supplementals.

3- It should be useful to integrate an extra figure just giving a brief overview of the created plasmids. The community should further validate the newly created resources rather than only the lab of the authors, so it would be helpful if you could briefly explain how to replace the fluorescent proteins with other target genes of interest. How would that be if there are three or four target genes? How would you ligate them together? Since the new genetic tools are a major part of the paper, it would further improve the manuscript.

Minor comments:

1- Line 15, Maybe the authors should explain that knock-in is an established technique to express two components at a similar level. Two identical genomic regions of chromosome 2 duplication in AX4 can be used to achieve almost the same level of expression. Highly efficient knock-in methods have been reported for Dictyostelium.

2- The authors state that combining the coding sequences of two proteins into a single transcript leads to notable decreases in expression levels with strain-dependent manner. To avoid confusion, it would be important to clarify if this phenomenon is common to various strains of Dictyostelium or just the specific difference between AX4 and NC28.1. These strains show originally many differences, including axenic or non-axenic strains, growth rates, drug sensitivity and mating types.

3- The authors create an interesting genetic toolkit, while its specific application is still ambiguous. Concerted expression of CRY2/CIBN is an interesting application, however quantitative control of protein levels would not be simple by P2A system. A slightly more extensive discussion of the applications specific to the system would be helpful.

4- Line 154, 90C  90°C

5- Line 160, 4C  4°C

6- Fig.3 legend, *: p¡0.05  *: p>0.05

6. PLOS authors have the option to publish the peer review history of their article (what does this mean?). If published, this will include your full peer review and any attached files.

Reviewer #1: **Yes: **Peggy Paschke

Reviewer #2: No

---

## [Author Response · Author response to Decision Letter 0]

20 Dec 2022

Please see our PDF "Response to Reviewers" for a formatted copy of our response to reviewers.

---

## [Decision Letter · Decision Letter 1]

18 Jan 2023

Self-cleaving 2A peptides allow for expression of multiple genes in *Dictyostelium discoideum*

PONE-D-22-08221R1

Dear Dr. Allyson E. Sgro,

We’re pleased to inform you that your manuscript has been judged scientifically suitable for publication and will be formally accepted for publication once it meets all outstanding technical requirements.

Kind regards,

Maria Gasset, Ph.D.

Academic Editor

PLOS ONE

Additional Editor Comments (optional):

Reviewers' comments:

Reviewer's Responses to Questions

**Comments to the Author**

1. If the authors have adequately addressed your comments raised in a previous round of review and you feel that this manuscript is now acceptable for publication, you may indicate that here to bypass the “Comments to the Author” section, enter your conflict of interest statement in the “Confidential to Editor” section, and submit your "Accept" recommendation.

Reviewer #1: All comments have been addressed

Reviewer #2: All comments have been addressed

2. Is the manuscript technically sound, and do the data support the conclusions?

Reviewer #1: Yes

Reviewer #2: Yes

3. Has the statistical analysis been performed appropriately and rigorously? 

Reviewer #1: Yes

Reviewer #2: Yes

4. Have the authors made all data underlying the findings in their manuscript fully available?

Reviewer #1: Yes

Reviewer #2: Yes

5. Is the manuscript presented in an intelligible fashion and written in standard English?

Reviewer #1: Yes

Reviewer #2: Yes

6. Review Comments to the Author

Reviewer #1: All points I made have been addressed.

Thanks also for answering my additional questions.

I am sure the plasmid will be of great use in future! Looking forward to give them a go.

Reviewer #2: The revised manuscript is convincing and the new genetic tools are valuable for Dictyostelium community. The paper is well written. I support its publication as it is.

7. PLOS authors have the option to publish the peer review history of their article (what does this mean?). If published, this will include your full peer review and any attached files.

Reviewer #1: **Yes: **Peggy Paschke

Reviewer #2: No

---

## [Editor Report · Acceptance letter]

21 Feb 2023

PONE-D-22-08221R1 

Self-cleaving peptides for expression of multiple genes in *Dictyostelium discoideum*

Dear Dr. Sgro:

I'm pleased to inform you that your manuscript has been deemed suitable for publication in PLOS ONE. Congratulations! Your manuscript is now with our production department. 

Kind regards, 

on behalf of

Dr. Maria Gasset 

Academic Editor

PLOS ONE